# Deterministic Component Mining for Multi-Framework UI2Code Generation

Zixiong Yang [* 1]   Linxiao Li [* 2]   Jiaye Lin [3]   Binrui Wu [4]   Xiaoyu Kang [5]   Jiechao Gao [† 6]

## Abstract

Automating User Interface (UI) generation substantially improves productivity and accelerates development by reducing engineering time and manual effort. Despite recent progress of Multimodal Large Language Models (MLLMs) in UI2Code, most existing approaches focus on a single HTML/CSS form and fail to systematically incorporate front-end frameworks such as React, Vue, and Angular. Moreover, their outputs are often verbose and hard to reuse at the component level. To address those issues, we propose Deterministic Component Mining (DCM), a multi-stage pipeline that couples MLLMs with a compact intermediate representation to enable multi-framework and component-oriented code generation. Firstly, a lightweight structure model predicts the representation of the DOM tree in JSON format, capturing the coarse layout from a webpage screenshot. Subsequently, we formulate deterministic rules to normalize the predicted DOM tree and mine reusable components with repetitive patterns via structural hashing and clustering, thereby yielding a portable intermediate representation. Finally, we employ a framework-conditioned prompting strategy governed by a binding specification and a file-block protocol to emit HTML/React/Vue/Angular code with explicit component props and repeat constructs. Extensive experiments demonstrate that DCM significantly outperforms baselines on automatic evaluation metrics and component-level reuse, while delivering consistent gains in multi-framework portability and overall code structural quality.

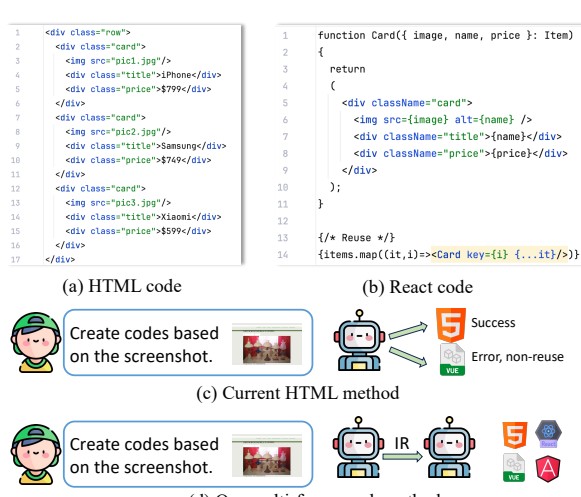

(a) HTML code          (b) React code

(c) Current HTML method

(d) Our multi-framework method

*Figure 1.* (a) HTML tends to duplicate structures and lacks reuse (Álvarez-Sabucedo et al., 2009); (b) React enables reuse via component abstraction; (c) Prior MLLM-based methods are HTML-only with limited reuse; and (d) Our method generates a framework-agnostic component-aware intermediate representation (IR), then conditions prompts to produce code across multiple frameworks.

## 1. Introduction

Automating the synthesis of User Interfaces (UIs) from designs can substantially improve developer productivity and shorten delivery cycles. This automation not only compresses development cycles but also enhances the consistency and maintainability of production interfaces at scale. Recent Multimodal Large Language Models (MLLMs) have demonstrated strong ability to translate UI designs into code (Kuang et al., 2025; Wang et al., 2025), opening new opportunities to replicate layout, text, and styling from screenshots with minimal manual effort (Wu et al., 2024).

Building on the rapid progress of MLLMs, several exploratory studies have been conducted, focusing on dataset creation and benchmarking. For instance, WebSight generates LLM-synthesized training pairs to scale up supervision for UI2Code (Vu et al., 2025). Design2Code (Si et al., 2025) curates a carefully collected test set and proposes an automatic metric for design-to-code similarity. WebCode2M (Gui et al., 2025) assembles a large real-world corpus for training and evaluation. Web2Code (Yun et al., 2024) provides a synthesized dataset and an MLLM-based evaluation

---
[*]Equal contribution [†]Corresponding author [1]Peking University [2]The University of Sydney [3]Tsinghua University [4]Fudan University [5]Chinese Academy of Sciences [6]Stanford University. Correspondence to: Jiechao Gao <jiechao@stanford.edu>.

framework. Beyond dataset curation, recent methodological advancements have sought to enhance generation fidelity through architectural innovations. ScreenCoder (Jiang et al., 2025) introduces a modular multi-agent framework to decouple grounding and planning for pixel-perfect replication (Li et al., 2026a), while LayoutCoder (Wu et al., 2025b) leverages explicit layout parsing trees to guide the structured fusion of code snippets for complex web structures.

However, existing works largely focus on HTML/CSS generation and overlook the component-based frameworks used in real-world web systems (e.g., React, Vue, Angular). The generated HTML/CSS codes typically exhibit extensive repetition rather than reusable components, inflating sequence length and maintenance overhead (see Fig. 1(a)). Compared with bare-metal HTML/CSS, MLLMs underperform in framework-centric development: they struggle with framework-specific syntax such as JSX parsing in React, template directives in Vue, and TypeScript-oriented patterns in Angular (Xiao et al., 2025). Consequently, how to leverage MLLMs for multi-framework UI2code generation while enhancing code reuse remains an open and under-explored problem (Yang et al., 2025; Xiao et al., 2025).

To address the above issues, we propose Deterministic Component Mining (DCM), a three-stage pipeline designed for multi-framework deployment and systematic component reuse. In the first stage, we train a lightweight structure model to predict a coarse-grained DOM tree from a webpage screenshot (represented in JSON), retaining only node types and hierarchy to obtain a stable structural skeleton. In the second stage, we develop rule-based normalization for the predicted DOM tree and leverage structural hashing and clustering to mine reusable components and repeat patterns under the same parent container. Meanwhile, we align cross-instance differences to identify variable content and generate slots, resulting in a portable intermediate representation. In the third stage, we adopt a conditioned prompting strategy to translate the intermediate representation into target framework code such as React, Vue, and Angular, enabling framework-consistent structure generation and component-level reuse. This design makes reuse semantics (component/repeat/slot) an interpretable and reproducible intermediate artifact, thereby mitigating the instability caused by long code generation and providing consistent structural constraints for multi-framework transfer (Gupta & Govil, 2010; Li et al., 2026b). Our main contributions are summarized as follows:

- We present the first systematic study of UI2Code generation and transfer across multiple front-end frameworks, including HTML, React, Vue, and Angular. We further propose a multi-stage pipeline centered on a portable intermediate representation, enabling the same UI structure to be robustly mapped to different implementations.

- We propose Deterministic Component Mining (DCM) that discovers reusable components and repeat patterns via rule-based design, and identifies slots through cross-instance difference alignment. This explicitly models reuse semantics, reducing redundancy while improving interpretability and reproducibility.

- We conduct extensive experiments along the dimensions of multi-framework generation and component reuse. Results show that DCM consistently outperforms baselines in automatic evaluation metrics and reuse quality.

## 2. Related Work

### 2.1. Multimodal Large Language Models

Multimodal Large Language Models (MLLMs)(Wu et al., 2025a; Li et al., 2025a), such as GPT-4o (Hurst et al., 2024), have revolutionized visual understanding across many domains (Li et al., 2025b; 2026d; Fu et al., 2026; Li et al., 2026c; Xu et al., 2026a;b). Unlike early CNN-LSTM architectures (Zhang et al., 2019), MLLMs leverage large-scale pre-training to perform zero-shot translation of UI screenshots, achieving promising results on established benchmarks like Design2Code (Si et al., 2025). However, recent studies highlight critical limitations in professional software engineering contexts. DesignBench (Xiao et al., 2025) notes that MLLMs excel at single-file HTML generation but consistently fail on modern component-based frameworks. Additionally, ScreenCoder (Jiang et al., 2025) and LayoutCoder (Wu et al., 2025b) demonstrate that MLLMs suffer from perceptual errors and spatial hallucinations, struggling to cleanly separate visual foundations from layout logic. These findings indicate that the inherently probabilistic nature of MLLMs is ill-suited for strict syntactic constraints (Qiu et al., 2026b), motivating the need for neuro-symbolic approaches (Zhang et al., 2025; Lai et al., 2025).

### 2.2. UI2Code Generation

**From Heuristics to Deep Learning** Early approaches like REMAUI (Nguyen & Csallner, 2015) relied on OCR and hand-crafted heuristics but failed to generalize reliably to diverse, real-world layouts. Pix2Code (Beltramelli, 2018) and Sketch2Code (Jain et al., 2019) then introduced deep learning to translate raw pixels directly into Domain-Specific Languages (DSLs). While pioneering, these monolithic end-to-end models lacked sufficient semantic understanding and struggled with long contexts, often generating structurally chaotic and unmaintainable code.

**Agentic and Layout-Guided Systems** To mitigate the "spaghetti code" and hallucination issues of MLLMs (Lam et al., 2025), the field has increasingly moved towards structured, modular, agent-based workflows (Qiu et al.,

2026a). ScreenCoder (Jiang et al., 2025) utilizes a "Layout-Language" for grounding, while LayoutCoder (Wu et al., 2025b) and WebVIA (Xu et al., 2025) incorporate explicit layout trees to further enhance robustness.

**Positioning of DCM**  While current agentic methods significantly improve visual fidelity, they predominantly focus on pixel-perfect replication rather than practical engineering reuse (Lu et al., 2025). Most still generate monolithic code, ignoring the component-based architecture (e.g., distinct headers, sidebars) that is vital for modern development (Yuan et al., 2025; Bohra et al., 2025). In contrast, our DCM framework introduces a "mining-before-generation" paradigm, explicitly extracting reusable structural patterns to ensure maintainable and modular code (Park et al., 2025).

## 3. Methodology

### 3.1. Task Definition

Given a webpage screenshot $x$ and a target frontend framework identifier $f \in \{\texttt{React}, \texttt{Vue}, \texttt{Angular}, \texttt{Vanilla}\}$, our goal is to generate well-structured, runnable frontend code $C_f$ in the specified framework such that its rendered output faithfully matches the screenshot $x$ as closely as possible in both structural layout and visual appearance.

### 3.2. Overview

As shown in Figure 2, we propose *Deterministic Component Mining* (DCM), which decomposes UI2Code into a three-stage pipeline: (1) **Vision-to-Structure:** learning a coarse-grained DOM tree $T$ from the screenshot; (2) **Structure-to-Reuse:** explicitly mining repeated structures on the predicted DOM tree to induce components and repeat semantics, yielding a cross-framework portable Intermediate Representation (IR); (3) **Reuse-to-Code:** translating IR into `React`/`Vue`/`Angular` code via framework-conditioned prompting, enforcing non-unrolled repeats and preserving component boundaries. The process can be written as:

$$T = G_\theta(x), \quad R = \mathcal{U}(T), \quad C_f = \text{LLM}(R, f), \quad (1)$$

where $G_\theta$ denotes the structure model, and $\mathcal{U}$ is a fully deterministic operator responsible for component mining and repeat injection. With this explicit decoupling, cross-framework differences are largely reduced to a small set of syntactic mappings and template-driven generation, improving reusability and maintainability, and making the overall system more amenable to scalable deployment.

### 3.3. Coarse DOM Tree Generation

Given a webpage screenshot $x$ and a structure generation model $G : x \mapsto T$, we first produce a coarse-grained DOM tree $T$. This lightweight tree retains only the node types,

hierarchical relations, and node geometry (BBox), serving to capture the global structural skeleton and major layout of the page, and providing a clean, stable input for the subsequent deterministic component mining stage.

In the webpage screenshot setting, inputs often come with high resolution and extreme aspect ratios, where naive cropping or resizing can destroy key layout cues. We adopt Pix2Struct as the implementation of $G$, mainly because it uses a ViT-style patch representation together with an aspect-ratio-preserving scaling strategy, making it more robust to varying resolutions and aspect ratios, and naturally accommodating variable sequence lengths and resolutions.

Concretely, Pix2Struct first applies aspect-ratio-preserving scaling to the input, then partitions the image into a fixed-size patch sequence $\{P_1, \ldots, P_N\}$. A ViT encoder produces the corresponding visual hidden states:

$$\{h_1, \ldots, h_N\} = \text{Enc}_\theta(\{P_1, \ldots, P_N\}). \quad (2)$$

A Transformer decoder then performs autoregressive next-token prediction conditioned on these visual hidden states to progressively generate a structured output.

We formulate structure prediction as a conditional sequence generation problem in JSON-formatted text: given the input screenshot, the model outputs a JSON string that encodes the DOM hierarchy via nested structures, while attaching the corresponding BBox attributes to nodes. Accordingly, we serialize the tree $T$ into a token sequence $y(T) = (y_1, \ldots, y_L)$, and model it autoregressively as:

$$p_\theta(y(T) \mid x) = \prod_{t=1}^{L} p_\theta(y_t \mid y_{<t}, x). \quad (3)$$

At decoding step $t$, the decoder outputs a hidden state $s_t$ (determined jointly by the visual states and previously generated tokens), and applies a softmax to obtain the distribution over the next token, e.g., $\text{softmax}(W \cdot s_t)$.

To ensure that the subsequent deterministic component mining can directly consume the generated structure, we impose strict JSON syntax and field constraints during generation and parsing, guaranteeing that the output sequence can always be reliably parsed into a valid tree.

### 3.4. Deterministic Component Mining

We take the coarse DOM tree $T$ as input and output an upgraded intermediate representation $R$. This phase deterministically augments it with explicit reusable semantics—components, repeat, and data tables—without introducing any additional learning parameters. Formally,

$$R = \mathcal{U}(T) = (T', \mathcal{C}, \mathcal{D}),$$

where $T'$ denotes the augmented tree after injecting repeat semantics and component references, $\mathcal{C}$ is the set of ex-

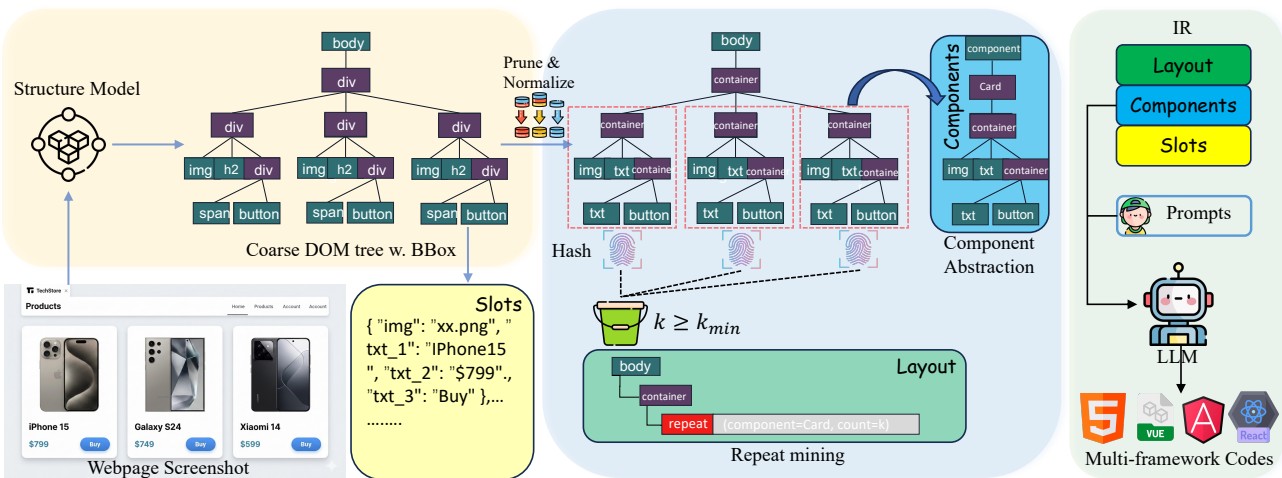

**Figure 2.** DCM decomposes UI2Code into three stages: structure generation, reuse mining, and multi-framework synthesis. By deterministically elevating repeated visual patterns into explicit IR semantics (components and repeats), framework differences are largely localized to the final prompt-based code synthesis stage, enabling consistent and portable generation across frameworks.

tracted component templates, and $\mathcal{D}$ is the collection of data tables associated with components and repeats.

**Canonicalization of DOM trees** To mine reusable patterns, the matching representation must be robust to contents (e.g., texts, hyperlinks, images), while the final code synthesis still needs to faithfully recover them. We therefore define a canonicalization method that transforms the DOM tree $T$ into two complementary views: structure view $\tilde{T}$ and content map $\mathcal{M}$. The structure view $\tilde{T}$ is used for fingerprinting and clustering: node tags are mapped to a stable vocabulary $\mathcal{K} = \{container, text, image, input, button, link\}$, sibling orders are canonicalized (e.g., by reading order or layout-aware ordering), and literal contents are masked by placeholders (e.g., TEXT/IMG/LINK). In parallel, the content map $\mathcal{M}$ stores generation-relevant attributes (e.g., text, href/src/alt, form values) indexed by canonical paths, ensuring that no necessary content is lost during canonicalization.

**Fingerprint of DOM subtrees** To efficiently discover structurally repeated patterns on the structure view $\tilde{T}$, we adopt a bottom-up structural fingerprint to compactly characterize DOM subtrees. Formally, this fingerprinting procedure is equivalent to a Merkle-hashing / Weisfeiler–Lehman (WL)-style recursion on ordered trees. We first define the canonical label of a node $v$ in $\tilde{T}$ as:

$$\ell(v) = \big(t(v), \operatorname{role}(v)\big), \qquad (4)$$

where $t(v)$ denotes the node type in $\tilde{T}$ and $\operatorname{role}(v)$ encodes the masked content role of $v$. This design deliberately excludes geometry and style attributes, retaining only coarse structural semantic categories, which makes the resulting fingerprints highly robust to noise.

Based on $\ell(\cdot)$, we compute a subtree hash $h(v)$ for each node $v$. Let the children of $v$ under a canonical order be $(c_1, \ldots, c_m)$. We adopt a unified reading order to ensure that isomorphic subtrees admit consistent serializations. We define the fingerprint of the subtree rooted at $v$:

$$h(v) = H\Big(\ell(v) \parallel h(c_1) \parallel \cdots \parallel h(c_m)\Big), \qquad (5)$$

where $H(\cdot)$ is a stable hash function and $\parallel$ denotes concatenation. This recursion guarantees that if two subtrees share exactly the same ordered parent–child relations and the same sequence of node labels under $\ell(\cdot)$, then their root hashes must be identical. In particular, for any parent node $p$, we can compute the multiset $\{h(c) \mid c \in \operatorname{ch}(p)\}$ of its children's hashes and bucket them by hash value to efficiently identify fully repeated local patterns, which provides candidate clusters for subsequent repeat mining.

**Repeat Mining** Repetitions in real-world webpages typically emerge among multiple sibling subtrees under the same shared parent container (e.g., list items, product cards, navigation entries). We therefore adopt a local, same-parent mining strategy as the primary and efficient mechanism. For any parent node $p$, let its children be $\operatorname{ch}(p) = \{v_1, \ldots, v_m\}$. We first bucket these children by their exact subtree fingerprints $h(v_i)$, yielding a set of hash-grouped clusters $\Gamma = \{\gamma_1, \gamma_2, \ldots\}$. For each cluster $\gamma$, if $|\gamma| \geq k_{\min}$ (e.g., $k_{\min} = 3$), we treat it as a candidate repeated pattern.

Given a repeated cluster $\gamma = \{u_1, \ldots, u_k\}$, we select a representative instance $u^\star$ as the component template (e.g., choosing the median-sized subtree to avoid extreme outlier noise). We then leverage the canonicalization outputs $(\tilde{T}, \mathcal{M})$ to accomplish two goals simultaneously, i.e., (1) Component template extraction: we rewrite the structure

view of $u^\star$ into a reusable component structure $C \in \mathcal{C}$; (2) Slot and data table generation: we align instances within the cluster and use $\mathcal{M}$ to identify varying positions, producing a slot set $\Sigma(C)$ and a data table $\mathcal{D}(C)$.

Finally, within the child sequence of $p$, we replace the original $k$ repeated instances with a single compact repeat node

$$\texttt{repeat}(C, \mathcal{D}(C)), \tag{6}$$

obtaining the upgraded tree $T'$. In our intermediate representation, $\texttt{repeat}$ explicitly carries references to the component $C$ and its data table $\mathcal{D}(C)$, making it a self-contained, portable loop semantic unit that can be consistently translated across different front-end frameworks.

**Nested Repeat**  Nested repetitions are common in practice (e.g., a "card list" where each card further contains a "tag chips" list). Without proper control, nested repeats may lead to overly fragmented and redundant components, degrading code readability and harming the stability of Stage 3 synthesis. We therefore perform explicit conflict resolution and greedy selection over all candidate repeat blocks.

For each candidate repeat block $\gamma$, we define a priority score

$$\text{score}(\gamma) = |\gamma| \cdot |V(u^\star)|, \tag{7}$$

where $|\gamma|$ denotes the repetition count and $|V(u^\star)|$ is the number of nodes in the template subtree (reflecting structural complexity). We then greedily select candidates in strictly descending order of $\text{score}(\cdot)$: once a repeat block is selected, we disallow selecting any smaller candidates whose covered regions fall inside it. This strategy prioritizes larger, more beneficial reusable units and avoids unnecessarily decomposing a single card into multiple fine-grained components. Despite its simplicity, the procedure is deterministic and reproducible, and it substantially improves the stability of component granularity in practice.

### 3.5. Prompt-Based Multi-framework Code Synthesis

This stage feeds the upgraded IR $R = (T', \mathcal{C}, \mathcal{D})$ together with the target framework $f$ into an LLM to synthesize runnable code $C_f$. We emphasize that the multi-framework capability primarily stems from the framework-agnostic nature of the IR, i.e., $\texttt{repeat/component/slot/data}$ encode abstract semantics, while the differences in generated code mainly lie in framework-specific syntax and engineering scaffolding. Specifically, we formulate code synthesis as a conditional generation problem:

$$C_f = \text{LLM}(\text{Prompt}(R, f)). \tag{8}$$

To reduce hallucinations and ensure executability, we impose four hard-constraint-style generation rules in the prompt: (1) every $\texttt{repeat}$ in the IR must be translated

into the native looping construct of the target framework (React: $\texttt{array.map}$, Vue: $\texttt{v-for}$, Angular: $\texttt{*ngFor}$); (2) component boundaries must be strictly consistent with $\mathcal{C}$, and each component's props must fully cover the slot set $\Sigma(C)$; (3) every field appearing in the data table $\mathcal{D}(C)$ must be explicitly consumed in rendering to prevent silent field dropping; (4) the output should minimize the number of files while remaining directly runnable (e.g., a single-file React component or a Vue SFC; for Angular, output the core component template and class definition).

Although the structure view $\tilde{T}$ used in Stage 2 masks literal contents during matching, the IR that enters the prompt fully retains $\mathcal{D}$ and the slot-binding relations. Therefore, texts, hyperlinks, and image resources are explicitly provided in structured data form. The LLM is no longer responsible for "guessing content from the screenshot"; instead, it binds $\mathcal{D}$ to component props via slots and renders them accordingly, which substantially reduces generation difficulty while improving output stability and reusability.

## 4. Experiments

### 4.1. Experimental Setup

**Datasets**  For training, we utilize two datasets, WebSight v0.1 (Laurençon et al., 2024) and WebCode2M (Gui et al., 2025), to obtain the structure model, each demonstrating distinct characteristics across various indicators. Compared with the WebSight dataset, WebCode2M's data is more complex, possesses a richer variety of styles and tag types, and is significantly longer, making it closer to real-world HTML code. Furthermore, WebCode2M provides the page's BBox information directly, which is essential for the training of our structure model. For the WebSight dataset, we also extract the BBox information prior to training. For evaluation, we evaluate all methods on the WebCode2M test datasets. WebCode2M test datasets are composed of three subsets: WebCode2M-Short, WebCode2M-Mid, and WebCode2M-Long. These subsets are created by splitting the data according to the length range of the ground-truth HTML code. The length ranges of the ground-truth HTML code for these three subsets are [551, 2045], [2052,4085], and [4098,10990], respectively. Each subset contains 256 samples.

**Evaluation Metrics**  (1) CLIP Similarity (Radford et al., 2021). CLIP is a multi-modal model trained using a contrastive objective on a large dataset of millions of internet textimage pairs. It learns to align images with their textual descriptions in a shared representation space. The latent vectors generated by the CLIP model capture the semantic information of the inputs. As a result, the cosine similarity between these vectors, which calculates the cosine of the angle between them, effectively quantifies the degree of similarity between the images. (2) Visual Score (Si et al.,

*Table 1.* Comparison of main benchmark results across different models grouped by target frameworks on WebCode2M. The best and the second-best results are highlighted in **bold** and underlined, respectively. Note that CSR and RR metrics are not applicable (-) to Vanilla HTML due to its interpreted nature and inherent lack of explicit component structure.

| Method | WebCode2M-Short | | | | WebCode2M-Mid | | | | WebCode2M-Long | | | |
|---|---|---|---|---|---|---|---|---|---|---|---|---|
| | CLIP | Visual | CSR | RR | CLIP | Visual | CSR | RR | CLIP | Visual | CSR | RR |
| *Vanilla HTML* | | | | | | | | | | | | |
| Gemini-2.5-Pro | 0.84 | 0.75 | - | - | 0.82 | 0.71 | - | - | **0.80** | 0.67 | - | - |
| GPT-4o | 0.82 | 0.73 | - | - | 0.79 | 0.69 | - | - | 0.74 | 0.65 | - | - |
| LLaVA-v1.5-7B | 0.68 | 0.43 | - | - | 0.65 | 0.40 | - | - | 0.62 | 0.35 | - | - |
| Qwen2.5-VL-7B | 0.80 | 0.64 | - | - | 0.76 | 0.60 | - | - | 0.72 | 0.56 | - | - |
| **DCM (Ours)** | **0.87** | **0.82** | - | - | **0.83** | **0.78** | - | - | 0.78 | **0.74** | - | - |
| *React Framework* | | | | | | | | | | | | |
| Gemini-2.5-Pro | 0.82 | 0.72 | 0.88 | 0.82 | 0.80 | 0.68 | 0.84 | 0.78 | **0.78** | 0.64 | **0.74** | **0.68** |
| GPT-4o | 0.81 | 0.70 | 0.86 | 0.80 | 0.79 | 0.67 | 0.81 | 0.75 | 0.72 | 0.62 | 0.70 | 0.64 |
| LLaVA-v1.5-7B | 0.65 | 0.40 | 0.55 | 0.49 | 0.63 | 0.38 | 0.48 | 0.42 | 0.60 | 0.33 | 0.40 | 0.35 |
| Qwen2.5-VL-7B | 0.78 | 0.61 | 0.72 | 0.65 | 0.74 | 0.58 | 0.66 | 0.59 | 0.70 | 0.54 | 0.58 | 0.52 |
| **DCM (Ours)** | **0.85** | **0.79** | **0.91** | **0.85** | **0.81** | **0.75** | **0.86** | **0.80** | 0.76 | **0.71** | 0.73 | 0.67 |
| *Vue Framework* | | | | | | | | | | | | |
| Gemini-2.5-Pro | **0.79** | 0.67 | 0.84 | 0.78 | 0.77 | 0.64 | 0.79 | 0.73 | **0.75** | 0.60 | 0.69 | 0.63 |
| GPT-4o | 0.77 | 0.66 | 0.82 | 0.76 | 0.76 | 0.63 | 0.77 | 0.71 | 0.69 | 0.59 | 0.66 | 0.60 |
| LLaVA-v1.5-7B | 0.62 | 0.37 | 0.52 | 0.46 | 0.60 | 0.34 | 0.45 | 0.39 | 0.58 | 0.30 | 0.36 | 0.30 |
| Qwen2.5-VL-7B | 0.75 | 0.57 | 0.65 | 0.58 | 0.70 | 0.53 | 0.59 | 0.52 | 0.67 | 0.49 | 0.52 | 0.46 |
| **DCM (Ours)** | **0.79** | **0.74** | **0.87** | **0.81** | **0.79** | **0.70** | **0.82** | **0.76** | 0.72 | **0.66** | **0.70** | **0.64** |
| *Angular Framework* | | | | | | | | | | | | |
| Gemini-2.5-Pro | 0.74 | 0.62 | 0.74 | 0.68 | **0.72** | 0.58 | 0.70 | 0.64 | **0.70** | 0.55 | **0.62** | **0.56** |
| GPT-4o | 0.72 | 0.60 | 0.71 | 0.65 | 0.70 | 0.56 | 0.67 | 0.61 | 0.64 | 0.52 | 0.58 | 0.52 |
| LLaVA-v1.5-7B | 0.58 | 0.31 | 0.42 | 0.36 | 0.56 | 0.29 | 0.35 | 0.28 | 0.54 | 0.26 | 0.25 | 0.20 |
| Qwen2.5-VL-7B | 0.69 | 0.51 | 0.54 | 0.48 | 0.65 | 0.48 | 0.49 | 0.43 | 0.62 | 0.44 | 0.41 | 0.35 |
| **DCM (Ours)** | **0.76** | **0.68** | **0.76** | **0.70** | **0.72** | **0.64** | **0.72** | **0.66** | 0.68 | **0.60** | 0.61 | 0.55 |

2025). It is used to assess the degree of alignment between low-level elements based on their appearance. The scores primarily evaluate the match ratio between reference and candidate blocks, as well as their similarity at the block level, considering factors such as color, text, and position. (3) Compilation Success Rate (CSR) (Xiao et al., 2025). This metric represents the percentage of generated code that compiles successfully without any framework-specific errors. Given that the total number of samples is $N$ and the number of successfully compiled samples is $S$, CSR $= \frac{S}{N}$. (4) Reusability Rate (RR): This metric evaluates the extent of component abstraction in the generated code. It represents the percentage of samples where the model successfully identifies repeated visual patterns and implements them using reusable component structures.

**Model Setup** We benchmark our approach against a comprehensive suite of state-of-the-art models spanning different scales and paradigms. This includes leading proprietary MLLMs, Gemini-2.5-Pro and GPT-4o; as well as a range of open-source MLLMs (LLaVA-v1.5-7B and Qwen2.5-VL-

7B). For all baseline models, we utilize the same standardized prompt for generation that is shown in Appendix A.

### 4.2. Main Results

**Overall Performance.** Table 1 summarizes the main results on WebCode2M under four target frameworks. Overall, DCM achieves consistently strong performance across WebCode2M-Short/Mid/Long, and it is particularly effective in preserving fine-grained low-level visual fidelity and producing executable, reusable implementations. While leading end-to-end MLLMs (e.g., GPT-4o and Gemini-2.5-Pro) can be competitive when generating Vanilla HTML, their performance becomes markedly less reliable when the target shifts to component-based frameworks (React/Vue/Angular), where the output must satisfy strict framework-specific syntax, component boundaries, and loop semantics. In contrast, DCM maintains high-quality results across all frameworks by converting the coarse DOM tree into a framework-agnostic IR with explicit `repeat` and component abstractions, and then synthesizing framework-specific code under these structural constraints.

Across frameworks and splits, DCM yields the highest (or near-highest) Visual scores, indicating consistently better alignment of layout and element-level appearance with the reference pages. For example, in the React setting, DCM improves the overall Visual score substantially over the strongest baseline on both Short and Mid splits, and it remains clearly better on Long even when the CLIP similarity becomes comparable across top methods. This pattern suggests that the deterministic mining and structured IR help reduce subtle fine-grained discrepancies that are difficult for pure end-to-end generation to control.

In addition, DCM significantly strengthens and consolidates engineering-oriented quality for component-based frameworks. As shown by CSR and RR in Table 1, DCM attains the best or second-best compilation success and component reuse rates across React, Vue, and Angular, demonstrating that the mined components and `repeat` nodes reliably translate into stable component boundaries and loop-based implementations. Notably, on the WebCode2M-Long split where pages are most complex, DCM still preserves strong executability and reuse with only a modest gap to the best baseline in a few cases, highlighting its remarkable robustness on large, real-world webpages.

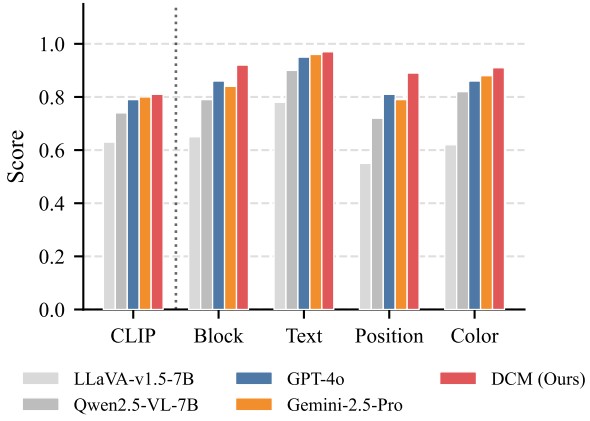

Figure 3. Detailed per-dimension sub-indicators of the overall visual score, including Block, Text, Position, and Color, for the React framework on the WebCode2M-Mid split.

**Visual Score Breakdown**  As illustrated in Figure 3, we perform a more fine-grained analysis of visual fidelity following the well-established evaluation taxonomy proposed in DesignBench (Xiao et al., 2025). The comparison reveals that DCM consistently outperforms both state-of-the-art proprietary models (GPT-4o, Gemini-2.5-Pro) and open-source baselines (LLaVA-v1.5-7B, Qwen2.5-VL-7B) across all five distinct dimensions: CLIP semantic alignment, Block layout, Text content, Element Position, and Color accuracy.

Notably, DCM achieves a substantial lead in the Position

and Block metrics. This empirical evidence supports our hypothesis that separating structure prediction (Stage 1) from code generation effectively and systematically mitigates the "spatial hallucination" often observed in end-to-end MLLMs, allowing for significantly more precise element placement. Furthermore, DCM attains near-perfect scores in the Text dimension. Unlike general MLLMs that generate text as probabilistic tokens—often leading to typos or hallucinations—our method treats text as structured data fields ($D$) within the structured Intermediate Representation, ensuring accurate, pixel-perfect content reproduction.

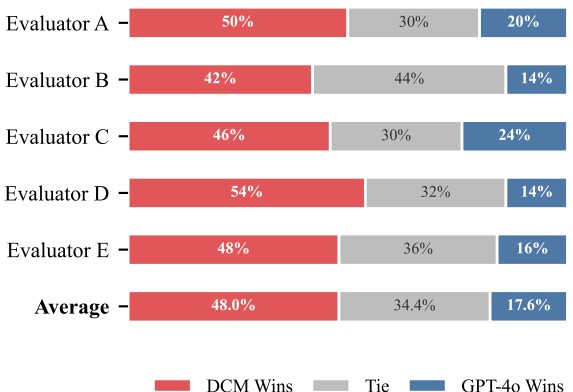

Figure 4. Human evaluation results from a blind pairwise comparison between DCM and GPT-4o, rated by five professional annotators based on visual faithfulness and code structural quality. Each bar represents the proportion of *Win*, *Tie*, and *Loss* cases.

**Human Evaluation**  As illustrated in Figure 4, we conduct a blind pairwise comparison between DCM and GPT-4o, to assess the overall generation quality from a human perspective. Five professional evaluators independently judge the outputs based on visual faithfulness and code structure. The results demonstrate a distinct preference for our method: on average, DCM is strongly favored in 48.0% of the cases, significantly outperforming GPT-4o, which achieves a win rate of only 17.6%. A substantial portion (34.4%) of cases are rated as ties, indicating broadly comparable visual quality in simpler layouts. However, the high win-to-loss ratio (approximately 2.7 : 1) confirms that for complex, real-world multi-framework scenarios, human experts consistently find DCM's component-aware code to be clearly superior in terms of maintainability and structural correctness.

### 4.3. Case Study

As illustrated in Figure 5, we present a qualitative comparison to demonstrate the end-to-end generation fidelity in diverse real-world scenarios. The example features a webpage header containing specific branding elements (logos) and richly structured text. While GPT-4o (Fig. 5(b))

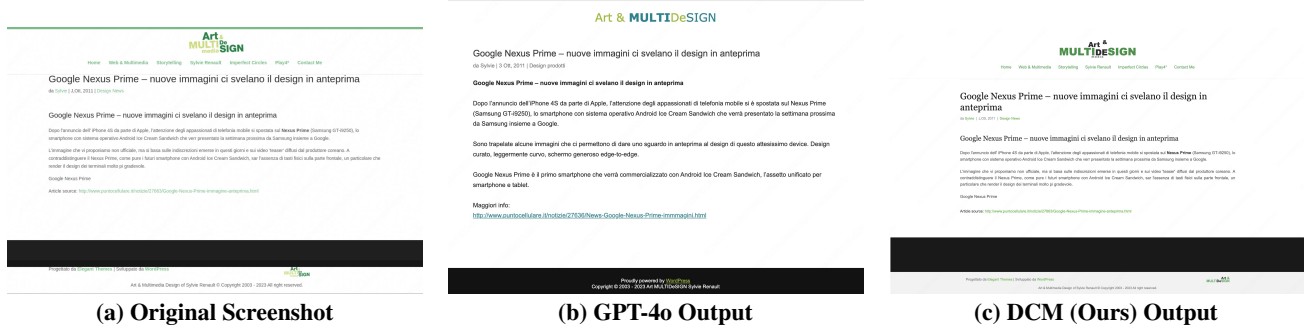

(a) Original Screenshot    (b) GPT-4o Output    (c) DCM (Ours) Output

*Figure 5.* Qualitative comparison of generated webpages: (a) the original reference screenshot, (b) the output of GPT-4o, and (c) the output of our DCM. GPT-4o exhibits visual artifacts including hallucinated logo text and spatial misalignment in the navigation bar, while DCM reproduces both the structural layout and textual content with pixel-level fidelity.

successfully retrieves the overall high-level layout, it still exhibits noticeable generative artifacts; specifically, it struggles with fine-grained text rendering (e.g., hallucinating or distorting the logo text) and introduces minor but observable spatial misalignments in the navigation bar. In contrast, DCM (Fig. 5(c)) produces a near-perfect, virtually indistinguishable replica of the original screenshot (Fig. 5(a)). By jointly leveraging our deterministic component mining and explicit data slot binding, DCM avoids the inherent probabilistic noise present in end-to-end MLLMs, ensuring that both the structural skeleton and the precise textual content are reproduced with pixel-level precision.

## 5. Conclusion

In this paper, we presented Deterministic Component Mining (DCM), a novel neuro-symbolic framework that bridges the critical gap between visual UI designs and engineering-ready frontend code. Unlike existing MLLM-based approaches that predominantly focus on monolithic HTML generation, DCM introduces a "mining-before-generation" paradigm. By decoupling visual structure prediction from code synthesis through a portable Intermediate Representation (IR), our method explicitly captures reuse semantics—including components, repetitive patterns, and data slots—in a fully framework-agnostic manner. Extensive experiments on the WebCode2M benchmark demonstrate that DCM significantly outperforms state-of-the-art multimodal LLMs, including GPT-4o and Gemini-2.5 Pro, across multiple mainstream front-end frameworks (React, Vue, and Angular). DCM not only achieves higher visual fidelity and compilation success rates but also produces modular, maintainable code with explicit, systematic component abstraction. Our work highlights the importance of hybrid approaches that combine the perceptual strength of vision encoders with the logical rigor of deterministic algorithms for complex, real-world software engineering tasks. In future work, we plan to extend DCM to support dynamic state management and complex interactive logic, further advancing the end-to-end automation of full-stack front-end development.

## Impact Statement

This paper aims to advance machine learning and automated front-end engineering by enabling multi-framework UI2Code generation with improved fine-grained component-level reuse. By separating structure prediction, deterministic component mining, and framework-conditioned code synthesis, the proposed approach can reduce manual engineering effort, significantly shorten prototyping cycles, and improve maintainability through explicit reusable components and repeat constructs. If deployed responsibly, it may benefit developers and organizations that need consistent UI implementations across heterogeneous technology stacks, and it can also support education by making UI development workflows substantially more accessible.

At the same time, UI synthesis technology can be misused to produce deceptive or fraudulent web interfaces at an unprecedented scale, including phishing pages or misleading replicas of real services. Generated code may also introduce security vulnerabilities, violate licensing constraints, or propagate accessibility and design issues present in training or evaluation data. These risks are not unique to our method, but they are relevant to any system that lowers the cost of producing realistic web interfaces. We encourage using such systems within well-established software engineering safeguards, including human review, security scanning, dependency and license compliance checks, and deployment controls. The intermediate representation and deterministic mining steps in our pipeline can facilitate auditing by making reuse structure and data bindings explicit, but they do not remove the need for responsible use and verification in downstream applications. We therefore call on practitioners and institutions to engage with emerging governance frameworks for software generation tools.

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

## A. Prompt

---

**Prompt for Code Generation**

**Task:** You are a code generator that transforms an abstract Intermediate Representation (IR) into executable frontend code. You will receive:

1. A Portable IR structure containing layout tree, components, and data.

2. A target framework specification (Vanilla HTML, React, Vue, or Angular).

**Input IR Structure:** The IR consists of three parts: $R = (T', C, D)$

- **T'**: Layout tree with repeat semantics and component references.

- **C**: Set of component templates with slot definitions.

- **D**: Data tables associated with components (slot $\rightarrow$ value mappings).

**Hard Constraints (MUST Follow):**

*Constraint 1: Loop Translation*
Every `repeat` node MUST be translated to the target framework's native loop construct:

- **React**: `{data.map((item, index) => <Component key={index} {...item} />)}`

- **Vue**: `<Component v-for="(item, index) in data" :key="index" v-bind="item" />`

- **Angular**: `<app-component *ngFor="let item of data; index as i" [item]="item"></app-component>`

- **Vanilla HTML**: Use JavaScript loop to generate HTML strings or DOM elements.

*Constraint 2: Component Props Coverage*

- Each component MUST accept props/attributes for ALL slots defined in $C$.

- Component boundaries MUST match the component definitions in $C$.

- Prop names MUST correspond exactly to slot names: $\Sigma(C) = \{slot_1, slot_2, ...\}$.

*Constraint 3: Data Field Consumption*

- Every field in data table $D(C)$ MUST be rendered in the output.

- No data field should be dropped or ignored.

- Validate that all slots have corresponding data bindings.

*Constraint 4: Single-File Output* Generate minimal, directly runnable code:

- **React**: Single JSX/TSX file with all components and data embedded.

- **Vue**: Single-File Component (.vue) with template, script, and style.

- **Angular**: Component template + class definition (2 code blocks max).

- **Vanilla HTML**: Single HTML file with embedded CSS and JavaScript.

**Output Format:** Provide the code in markdown code blocks with appropriate language tags.

---

**Now generate code for the following IR and target framework:**
**Target Framework:** [Vanilla HTML / React / Vue / Angular]
**IR:** [PASTE IR HERE]

---

# B. Case Study

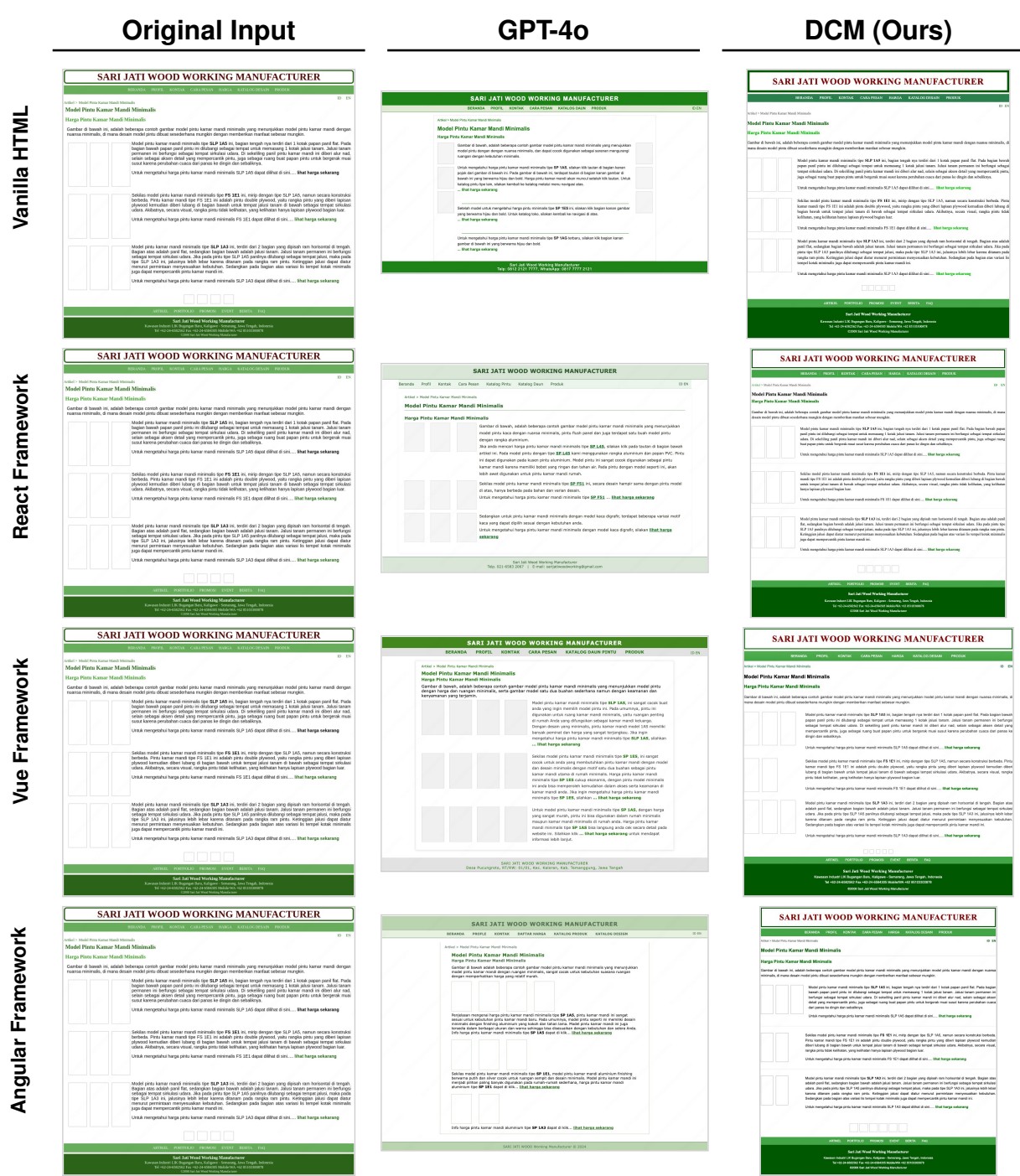

*Figure 6.* Qualitative comparison across frameworks. We present the original input screenshot (left), the output from GPT-4o (middle), and the output from our DCM (right). The gray borders indicate the viewport boundaries.

## C. Additional Methodology Details

### C.1. Stage 3 as Constrained Syntax Realization

A potential concern about Stage 3 is whether it merely wraps an LLM with prompt engineering and therefore inherits the unreliability of free-form generation. We clarify here that Stage 3 is fundamentally different from unconstrained generation. All structural reasoning, i.e., component boundary identification, repeat pattern mining, slot extraction, and data table construction, is completed deterministically in Stages 1 and 2 before the LLM is ever invoked. The intermediate representation $R = (T', \mathcal{C}, \mathcal{D})$ that enters Stage 3 fully specifies:

- which subtrees are repeated and how many times (`repeat` nodes in $T'$);

- which subtrees form reusable components and what their slot interfaces are ($\mathcal{C}, \Sigma(C)$);

- what concrete data values fill each slot for each instance ($\mathcal{D}(C)$).

The LLM is therefore not asked to infer structural decisions or component boundaries from pixels. Its role is constrained syntax realization: given a fully specified structural IR and a target framework, translate each abstract IR node into the corresponding framework-specific syntax under deterministic hard constraints (loop translation, component props coverage, data field consumption, single-file output). The multi-framework capability is a direct consequence of the IR's framework-agnostic design: the same `repeat`/`component`/`slot`/`data` IR maps to `array.map` in React, `v-for` in Vue, and `*ngFor` in Angular, with the LLM only responsible for the syntactic realization. This architectural separation ensures that structural correctness is enforced deterministically and is independent of the LLM's generative tendencies.

### C.2. Stage 2 Robustness to BBox Noise

A natural question is whether Stage 2's component mining is brittle to BBox prediction errors from Stage 1. We show here that Stage 2 is inherently robust to BBox noise by design. The subtree hash $h(v)$ is computed from the node label $\ell(v) = (t(v), \mathrm{role}(v))$ and the recursively computed hashes of $v$'s children. Crucially, neither $t(v)$ nor $\mathrm{role}(v)$ involves any BBox coordinate: $t(v)$ is the semantic node type from the vocabulary $\mathcal{K}$ (container, text, image, input, button, link), and $\mathrm{role}(v)$ encodes the masked content category. Geometry and style attributes are deliberately excluded. Therefore, any BBox error from Stage 1, whether a position offset, a size inaccuracy, or a missed detection, has no effect on the subtree fingerprints or the repeat mining step. The BBox output of Stage 1 is consumed only at Stage 3, where it informs the rendering order of nodes in the final prompt (i.e., spatial layout guidance for the LLM). It plays no role in the fingerprinting, clustering, or IR construction steps of Stage 2. This architectural isolation means that Stage 2 provides a natural noise barrier: structural mining is fully decoupled from the geometric regression quality of Stage 1.

## D. Additional Experiment Details

### D.1. Dataset Characteristics and Evaluation Split Criteria

We use two datasets for training the Stage 1 structure model.

**WebSight v0.1** is an LLM-synthesized dataset of HTML/screenshot pairs. Its pages feature relatively simple layouts with a limited tag vocabulary, making it a useful source for learning broad structural patterns. Because WebSight does not provide BBox annotations directly, we extract per-element BBox information from the HTML DOM prior to training.

**WebCode2M** is a large-scale real-world corpus harvested from live websites. Compared to WebSight, its pages are more complex, with richer styles, greater diversity, and longer HTML, making them closer to production-grade web frontends. WebCode2M provides BBox information directly as part of each sample, which is essential for training the structure model.

These two datasets serve complementary roles: WebSight helps the model learn generalizable structural patterns in a controlled, clean-data setting, while WebCode2M exposes it to the full complexity and diversity of real-world web pages.

For evaluation, all methods are evaluated on the official WebCode2M test set. The test set is partitioned into three complexity subsets based solely on the length of the ground-truth HTML code: **Short** [551, 2045], **Mid** [2052, 4085], and **Long** [4098, 10990] tokens. Each subset contains 256 samples. This length-based split is designed to probe scalability under increasing page complexity (since longer HTML generally corresponds to more DOM nodes, more repeated structures, and more

complex visual layouts), rather than to introduce a semantic partition. Importantly, evaluation uses only the screenshot and BBox pairs as visual reference; the ground-truth HTML code is not used as supervision or as a direct generation target, so all four target frameworks (Vanilla HTML, React, Vue, Angular) are evaluated against the same visual reference standard.

### D.2. Reusability Rate (RR): Full Operationalization

The Reusability Rate (RR) is an end-to-end engineering metric applied to the final generated code, not to DCM's intermediate representation. It is defined and computed as follows:

- **Page Selection.** For each reference page in the evaluation set, we first determine whether it contains repeated visual structures that should naturally be implemented through reuse (e.g., repeated cards, list items, navigation entries, table rows). Pages without any qualifying repeated structure are excluded from the RR denominator.

- **Positive Judgment.** For each retained page, we inspect the generated React/Vue/Angular output and check whether the repeated visual structures are realized through (a) an actual reusable component definition (positive) or a framework-native loop construct (`array.map`, `v-for`, `*ngFor`) that iterates over a data array (positive), rather than (b) manually duplicated, page-specific code blocks (negative).

- **Score Computation.** RR is the fraction of positive samples among all retained samples for that framework.

Because Vanilla HTML lacks explicit component or loop semantics, RR is only reported for component-based frameworks. RR should be interpreted jointly with CLIP, Visual Score, and CSR rather than in isolation: a model could achieve high RR by always generating loops even when the page does not contain repetition, which would degrade other metrics.

### D.3. Angular CSR Analysis

The Compilation Success Rate (CSR) for Angular is consistently lower than for React and Vue across all methods in our benchmark, not only for DCM. This reflects an inherent difficulty of Angular as a generation target rather than a weakness specific to our approach. Angular imposes significantly stricter constraints than React or Vue: (1) its template syntax requires matching `[property]` and `(event)` bindings that are sensitive to exact naming and nesting; (2) TypeScript typing is enforced at compile time, meaning that type mismatches introduced by any generation system produce compile errors; (3) Angular's module system requires explicit declarations and imports that are difficult for end-to-end models to generate correctly. As a result, even state-of-the-art proprietary MLLMs such as GPT-4o produce lower CSR on Angular than on React or Vue. Importantly, DCM remains the strongest or near-strongest method even under this harder setting. On Angular Short/Mid/Long, DCM improves CSR over GPT-4o across splits. This demonstrates that the explicit IR and structural hard constraints provide meaningful gains for the most demanding framework target.

### D.4. Human Evaluation Protocol

We randomly sample 100 reference pages from the WebCode2M test set, stratified across Short, Mid, and Long complexity splits (approximately 33–34 pages per split) to ensure that all complexity levels are represented.

Five professional front-end developers serve as evaluators. Each evaluator is presented with side-by-side rendered outputs from DCM and GPT-4o for the same reference screenshot, without any label indicating which system produces which output (relative blind evaluation). The order of the two outputs (left/right) is randomized per sample.

Each evaluator independently rates each comparison as one of three options: DCM preferred, GPT-4o preferred, or Tie. Judgments are based on two equally weighted criteria: (1) Visual faithfulness: how closely the rendered output matches the reference screenshot in terms of layout, color, element placement, and text content; and (2) Code structure quality: whether the generated code exhibits good component decomposition, reusability, and maintainability.

Inter-annotator agreement across five evaluators is measured using Fleiss's $\kappa$. The resulting $\kappa = 0.63$ indicates substantial agreement according to the standard scale, confirming that the evaluation criteria are applied consistently across annotators.

A two-sided sign test is conducted to determine whether DCM's empirical win rate (48.0%) is significantly above the chance level (50% split between win and loss, excluding ties). Excluding ties, DCM wins 240 of the 328 non-tied judgments; the test yields $p < 0.001$, confirming that the observed preference for DCM is statistically significant.

