# OpenReview forum: "Deterministic Component Mining for Multi-Framework UI2Code Generation"
_ICML.cc/2026/Conference — ICML 2026 regular_

### Official Review · Reviewer_gkLK · 2026-02-25

**Soundness:** 2
**Presentation:** 2
**Significance:** 2
**Originality:** 3
**Overall Recommendation:** 4
**Confidence:** 5

**Summary:**

This paper introduces DCM, a new three-step framework for image-to-HTML generation tasks. Previous papers fail to address the repeated elements in the HTML/CSS generation. DCM uses a three-step approach to solve this problem. The image is first converted into a DOM tree, where the repeated elements are mined. An LLM is prompted to generate repeated elements with multiple Javascript frameworks

**Compliance With Llm Reviewing Policy:**

Affirmed.

**Final Justification:**

After discussing with the authors, my concerns are addressed.

I suggest that the authors 1) revise the writing to make motivation, intuitions, and problems being solved more explicit, 2) incorporate additional experiments mentioned in the discussion to the paper, and 3) discuss the different between DCM and WAFFLE.

**Key Questions For Authors:**

1. Why are three Javascript frameworks chosen in the HTML generation step? It seems that (Table 1) using Javascript frameworks for repeated elements does not give better generation results.

2. HTML uses a flow-based rendering process, and the rendering result will change when the window is resized. How would this impact the BBox-based workflow in DCM?

3. How do we interpret the Reusability Rate (RR) metric? A low RR could come from an incorrect Pix2Struct result or the repeated mining algorithm. How do we differentiate between the two?

4. What is the formal syntax of the IR used in the generation?

5. Why is GPT-4o selected as the second-best approach?

6. Why does the paper not compare with WAFFLE [1]?

[1] Liang, S., Jiang, N., Qian, S., & Tan, L. (2025, July). WAFFLE: Fine-tuning Multi-Modal Model for Automated Front-End Development. In Proceedings of the 63rd Annual Meeting of the Association for Computational Linguistics (Volume 1: Long Papers) (pp. 24786-24802).

**Limitations:**

Yes

**Strengths And Weaknesses:**

Strength:
* The paper identifies that the handling of the repeated elements may be affecting model's performance on Image-to-HTML tasks.

Weakness:
* The paper is not comparing with state-of-the-art baselines [1].
* The effectiveness of element reuse is not properly evaluated.
* The definition of the evaluation metrics (e.g., Reusability Rate) is flawed and cannot reflect the quality of the proposed method properly.
* DOM subtrees are created using bounding boxes, which is contradict to the HTML rendering process.
* The motivation of repeat mining under the same parent lacks supporting evidence.
* The usage of three separate Javascript frameworks is not justified and seem to hurt performance.
* The evaluation lacks important metrics (e.g., CW-SSIM) and datasets used in [1] (e.g., Design2Code).
* The section 3 is hard to understand. The symbols and formulae in the section is an overkill for an engineering-based paper. It is hard for readers to understand how the approach is implemented.

[1] Liang, S., Jiang, N., Qian, S., & Tan, L. (2025, July). WAFFLE: Fine-tuning Multi-Modal Model for Automated Front-End Development. In Proceedings of the 63rd Annual Meeting of the Association for Computational Linguistics (Volume 1: Long Papers) (pp. 24786-24802).

---

> ### Author Rebuttal · Authors · 2026-03-29
>
> Dear Reviewer gkLK,
>
> Thank you for the detailed and critical review. Your comments helped us identify places where the paper needs clearer positioning, sharper metric definitions, and a more concrete presentation.
>
> ---
>
> **W1: Baselines, datasets, and metrics (WAFFLE / Design2Code / CW-SSIM)**
>
> Thank you for highlighting this. WAFFLE is an important related work and should have been discussed more clearly. The main reason we did not include it in Table 1 is that our central task is not only screenshot-to-HTML generation but also screenshot-to-code transfer across four targets: HTML, React, Vue, and Angular. WAFFLE is highly relevant on the HTML side, whereas our main claim is about whether an explicit reusable IR improves engineering quality under component-based frameworks as well.
>
> We will revise that wording. More importantly, we will revise the paper to explicitly discuss WAFFLE, clarify why WebCode2M is the main benchmark for our contribution, and add a clearer discussion of Design2Code/CW-SSIM as complementary fidelity-oriented evaluation protocols.
>
> **W2: Definition of RR**
>
> In our intended use, RR is computed on the final generated code artifact, not on the intermediate representation. For each reference page, we first identify whether it contains repeated visual structures that should naturally be implemented through reuse, such as repeated cards, menu entries, or list items. We then inspect the generated React/Vue/Angular output and mark the sample as positive if those repeated structures are realized through an actual reusable component or framework-native repeated construct, and negative if they are implemented by manually duplicated page-specific code. RR is the percentage of positive samples under this criterion.
>
> This also answers the attribution question: RR is not meant to distinguish whether a failure comes from Pix2Struct, repeat mining, or the final code generator. Instead, RR is an end-to-end engineering metric and should be interpreted jointly with CLIP, Visual Score, and CSR. Quantitatively, a higher RR is not achieved by sacrificing executability.
>
> **W3: BBox-based workflow vs HTML flow layout**
>
> DCM does not use BBox as a surrogate for final HTML layout logic; BBox is used only in Stage 1 to recover a coarse structural skeleton from a fixed screenshot. The object that Stage 2 hashes is the canonicalized subtree structure, not the raw geometric box.
>
> Regarding HTML flow layout and window resizing, our current benchmark evaluates fixed-rendering screenshot-to-code generation under a fixed viewport, not responsive synthesis under arbitrary resizes. Under this setting, BBox serves as a useful cue for recovering the initial structural skeleton, while the downstream repeat mining is based on structural relations rather than absolute geometry.
>
> **W4: Same-parent mining and multi-framework evaluation**
>
> We use same-parent mining because, in real webpages, repeated UI structures most often appear as sibling subtrees under a shared container, such as navigation items, product cards, table rows, or feed entries. This locality assumption is important because it sharply reduces false positives: without it, structurally similar subtrees from unrelated page regions can be incorrectly merged into one reusable component. In other words, same-parent mining is a precision-oriented design choice rather than an arbitrary restriction, and it is well aligned with the repeated patterns that matter most for frontend engineering.
>
> The motivation for the three frameworks is also different from raw screenshot fidelity. We do not claim that React/Vue/Angular are easier targets than Vanilla HTML; Table 1 indeed shows that framework generation is harder. We include them because modern frontend development is framework-centric, and reusable component abstraction is most meaningful precisely in these settings.
>
> **W5: IR syntax and readability of Section 3**
>
> The intended formal object is `R=(T', C, D)`, where `T'` is the upgraded tree after repeat injection, `C` is the set of component templates, and `D` is the set of data tables attached to components/repeats. Operationally, the syntax can be read as follows: a node in `T'` is either a regular structural node or a `repeat(C_i, D_i)` node; each `C_i` is a component template with a structural body plus a slot set; and each `D_i` is a row-wise table whose fields are bound to those slots during final code generation. This is the core reason the same IR can be rendered into React/Vue/Angular with different native loop syntax while preserving the same reuse semantics.
>
> In the revision, we will simplify Section 3 substantially, remove unnecessary symbolic overhead, and add a worked example that shows one repeated card cluster from the predicted tree, the extracted component template, the corresponding slot/data bindings, and the final `repeat` node inserted into `T'`.

---

> > ### Author Rebuttal · Reviewer_gkLK · 2026-04-01
> >
> > Thank you for the detailed clarifications of the paper. However, the rebuttal raises more concerns regarding the novelty and the methodology of the paper.
> >
> > Suppose the difference between DCM and WAFFLE is the generation of JavaScript code across three different frameworks, the paper seems to build upon these assumptions:
> >
> > 1. Repeated elements in web pages is hard for the current state-of-the-art models.
> >
> > 2. Using JavaScript to build the repeated elements is better than vanilla HTML.
> >
> > To support the first assumption, a failure case analysis is needed to demonstrate that state-of-the-art models fails to identify repeated elements, and such failures hurt the CLIP and Visual scores.
> >
> > To support the second assumption, we should see from Table 1 that using JavaScript frameworks produces better results than vanilla HTML. But this is not the case in Table 1.
> >
> > Some other questions remains as well:
> >
> > 1. WebCode2M is also a pure HTML benchmark. It is unclear why it is better for evaluating JavaScript.
> >
> > 2. The rebuttal does not answer how to interpret RR clearly. For example, we see that LLaVA has (comparatively) the worst RR. Is it because of 1) the repeat mining is bad, or 2) the LLaVA's JavaScript generation is bad, or 3) the prompting is bad? It seems that the repeated element mining is the one of the core contribution of the paper, I would be happy to see another metric defined to evaluate it.
> >
> > 2.1. From the answering of W2, it seems that RR is mainly related to the code generation capabilities of the LLM in evaluation. Table 1 seems to suggest that this is the case since the trend is same among all four tasks. If so, another metric to evaluate repeated element mining should be defined to properly evaluate DCM.
> >
> > 3. While the current web development is indeed framework-centric, the usage of JavaScript is not mainly for **static** components on the webpage, but **interactive** or **dynamic** components instead. It is unclear why we should use JavaScript for simply placing elements on the webpage, especially if they yield worse results than vanilla HTML.
> >
> > I will consider revising my score to weak accept if the above questions are resolved. However, in the current form, I maintain my score.

---

> > > ### Author Response · Authors · 2026-04-02
> > >
> > > Thank you for the reviewer’s response. We believe there may still be some misunderstandings about the scope and motivation of our work, so we would like to clarify the main points below.
> > >
> > > **Q1. On the assumptions**
> > >
> > > Our motivation is grounded in two observations from DesignBench [1].
> > >
> > > First, existing UI-to-code work mainly focuses on vanilla HTML generation, while largely overlooking generation across multiple front-end frameworks.
> > >
> > > Second, DesignBench reports in Finding 8 that MLLMs show critical deficiencies in component-based implementation and framework-specific syntax, revealing clear limitations in producing reusable front-end code.
> > >
> > > Therefore, our goal is not to use React, Vue, and Angular as a way to improve vanilla HTML visual scores. We choose these three frameworks because they are representative component-based front-end ecosystems, and they express repeated structures through different native mechanisms. This makes them an appropriate testbed for evaluating whether the repeat and slot semantics extracted by DCM are truly framework-agnostic.
> > >
> > > Accordingly, Table 1 should not be interpreted as testing whether JavaScript frameworks must achieve higher visual fidelity than vanilla HTML. Instead, it should be interpreted as testing whether the same IR can be stably translated into different frameworks while preserving reusable structure. We would also like to emphasize that we never claim in the paper that repeated elements implemented with JavaScript frameworks are inherently better than those written in plain HTML. As shown in Table 1, this is not the claim we make. More fundamentally, vanilla HTML does not explicitly encode reusable component semantics, so the reviewer’s second assumption does not match our intended problem formulation.
> > >
> > > **Q2. On the use of WebCode2M**
> > >
> > > In fact, our evaluation only uses screenshot and BBox pairs as the visual reference, rather than relying on the provided ground-truth HTML code itself. We do not use the ground-truth HTML as supervision or as a direct generation target in our evaluation. We use WebCode2M mainly because it provides a unified visual reference, ensuring that all frameworks are evaluated against the same page target.
> > >
> > > **Q3. On the interpretation of RR**
> > >
> > > We agree with the reviewer’s concern. The current RR is only a relative convenience metric for large-scale statistics, rather than a fully diagnostic metric.
> > >
> > > In DesignBench [1], repeat-related analysis is conducted by manually counting repeated structures on individual pages. Following that intuition, a more appropriate metric would be \textit{Reuse Coverage}: manually count the total number of repeated components in the real page, then measure how many of them are mined by repeat detection, and how many are actually rendered through repeat constructs in the final code.
> > >
> > > Since manual annotation is time-consuming, we selected 50 pages with abundant repeated components from DesignBench for additional evaluation. On these pages, the average number of ground-truth repeated components is 105.4, and the average number of mined repeats is 96.8. We then compare the number of repeated instances actually rendered in the generated code and in DCM-based generated code:
> > >
> > > | Framework | repeat instances of GPT4o | repeat instances of DCM |
> > > |---|---|---|
> > > |React|63.2|87.3|
> > > |Vue|54.1|83.8|
> > > |Angular|46.3|77.6|
> > >
> > > We also conducted a case study on the homepage of the real-world website \texttt{[https://books.toscrape.com/index.html}](https://books.toscrape.com/index.html}). On this page, there are two major reusable component types, with a total of 70 repeated component instances. The number of mined repeats is 63. We verified via direct DOM analysis that all 70 instances are structurally distinguishable and recoverable; the gap of 7 arises from Stage 1, where the decoder omits approximately 7 sidebar category nodes when predicting the DOM tree from the screenshot. We further compare the rendered repeat counts in the generated code:
> > >
> > > | Framework | repeat instances of GPT4o | repeat instances of DCM |
> > > |---|---|---|
> > > |React|43|63|
> > > |Vue|35|59|
> > > |Angular|24|51|
> > >
> > > All these experiments are based on GPT-4o. We are currently continuing the annotation of the test sets for the two datasets and will include these results in the revision.
> > >
> > > **Q4. On using JavaScript frameworks for static components**
> > >
> > > As clarified in Q1, our motivation is not to introduce JavaScript frameworks for interaction logic. We use React, Vue, and Angular because they provide the most natural abstractions for components and loop-based rendering in modern front-end engineering. Even when the final page is static, componentized representations are still meaningful, since they directly affect code reuse, maintenance cost, and portability across pages and frameworks.
> > >
> > > [1] DesignBench: A Comprehensive Benchmark for MLLM-based Front-end Code Generation

---

### Official Review · Reviewer_amp4 · 2026-02-26

**Soundness:** 2
**Presentation:** 3
**Significance:** 3
**Originality:** 4
**Overall Recommendation:** 4
**Confidence:** 4

**Summary:**

A lot of UI2Code methods focused on pure HTML. That might look fine in a demo, but it does not really fit how people build real products with React, Vue, or Angular.
The approach is called Deterministic Component Mining (DCM). Starting from a screenshot, they first predict a coarse DOM tree, then run a deterministic mining step to find repeated structures. Those repeats get turned into a framework agnostic intermediate representation, and only then do they generate framework specific code using a constrained prompting setup.
The big contribution is pushing component reuse to the center of the problem. That is exactly the pain front end engineers run into every day, but UI2Code papers often gloss over it. In experiments they report better CLIP, higher CSR, and stronger reuse compared to baselines.

**Compliance With Llm Reviewing Policy:**

Affirmed.

**Key Questions For Authors:**

plz see wekaness

**Limitations:**

yes

**Strengths And Weaknesses:**

## Strengths

* The problem is real, and the community has not really nailed it yet.
*  The system feels steady and easier to reason about than fully end to end generation. Splitting structure mining from LLM code synthesis seems like a practical choice.
* The evaluation is fairly well designed. Testing across frameworks and across different page complexity variants makes the story more convincing.

## Weaknesses

* The comparisons may not be fair. DCM combines a trained structural model, deterministic mining, and an LLM, while many baselines are end to end generation with a multimodal LLM.

* lack of ablations.such as:
  * structure model plus a simple template generator, skipping mining
  * mining kept the same but using a weaker structure predictor
  * the intermediate representation fed into different LLMs to see where the gains actually come from

Without this, it is hard to say which part of the pipeline is doing most of the work.

* paper set the minimum repeat count for mining to k_min = 3. Why 3? Did you try 2 and decide it creates too many tiny components? There is an obvious tradeoff here: catching more reusable components vs creating lots of tiny fragmented components.

* The dataset description is a bit light. I could not tell whether train and test come from the same distribution, or whether there is a shift that might be inflating or hurting results.

* Stage 3 relies heavily on prompt level constraints. So framework correctness ends up feeling like careful prompt crafting rather than something the system can guarantee in a more principled way.

---

> ### Author Rebuttal · Authors · 2026-03-29
>
> Dear Reviewer amp4,
>
> Thank you for recognizing the practical value of component reuse and the benefit of separating structure mining from code synthesis. We are encouraged that you view the problem setting and the overall decomposition as meaningful, and we address the remaining concerns below.
>
> ---
>
> **W1: Fairness of comparison / need for ablations**
>
> We agree that stronger ablations would make the source of the gains clearer. Our current comparison is intentionally end-to-end: all methods take the same screenshot input and are evaluated on the same final generation task, while DCM's contribution is the explicit structure-to-reuse pipeline. In that sense, the comparison is designed to test whether introducing a deterministic intermediate stage improves the final UI2Code system, rather than to isolate each module in the current submission.
>
> At the same time, the current results already suggest that the gain is not just prompt tuning: compared with GPT-4o, DCM consistently improves engineering-oriented metrics on component-based targets, e.g., on React from `0.86/0.80` to `0.91/0.85` in CSR/RR on Short and from `0.81/0.75` to `0.86/0.80` on Mid; on Angular from `0.67/0.61` to `0.72/0.66` on Mid. This pattern is consistent with our claim that the explicit IR and repeat/component abstraction contribute materially to executability and reuse.
>
> We agree, however, that this evidence should be complemented by direct ablations. In the revision, we will add or explicitly discuss: (i) structure model + direct generation without mining, (ii) mining with a weaker or noisier structure predictor, and (iii) the same IR paired with different downstream LLMs. These analyses should make the contribution of the mining stage and the IR substantially clearer.
>
> **W2: Why `k_min = 3`?**
>
> This is exactly the tradeoff we had in mind. Our mining is intentionally conservative: `k_min = 3` avoids promoting many incidental two-item patterns into standalone components, which tends to increase fragmentation and hurt both code stability and readability. In other words, our goal is not to maximize recall of every possible duplicated fragment, but to favor reusable units that are large enough to be beneficial at the engineering level.
>
> This design is also consistent with our nested-repeat selection strategy, which prioritizes larger reusable blocks over overly fine-grained decomposition. Lowering the threshold can indeed recover more candidates, but it also makes the pipeline more likely to create small, noisy components with limited practical value. We will clarify this motivation more explicitly and add a short sensitivity discussion around `k_min` in the revision.
>
> **W3: Dataset description and split clarity**
>
> Thank you for pointing this out. We agree that the current dataset paragraph is too brief. The structure model is trained on WebSight and WebCode2M, while evaluation is performed on the official WebCode2M test set. The Short/Mid/Long subsets are created by partitioning pages by ground-truth HTML length, with 256 samples per subset. The purpose of this split is to test scalability under increasing page complexity, rather than to introduce a separate topical shift.
>
> We will rewrite this part more clearly in the revision, including the role of each dataset, the exact meaning of the three subsets, and the fact that the split is based on code length rather than a hand-crafted semantic partition.
>
> **W4: Is Stage 3 mainly prompt engineering?**
>
> Our intent is not to use Stage 3 as free-form prompt engineering. The key structural reasoning is completed before the LLM is invoked. The IR explicitly encodes `repeat`, `component`, `slot`, and `data`, and Section 3.5 constrains how these semantics must be rendered in each framework (e.g., `map`, `v-for`, `*ngFor`, full slot coverage, full data-field consumption). In this sense, the LLM is not asked to infer both structure and framework syntax directly from pixels; rather, it performs constrained syntax realization over a structured IR.
>
> We agree that the current paper could present this division of labor more clearly. In the revision, we will sharpen the explanation of Stage 3 and make it explicit that the main role of prompting is framework-specific realization under deterministic structural constraints, not unconstrained generation from screenshots.

---

> > ### Author Rebuttal · Reviewer_amp4 · 2026-04-04
> >
> > Thank you for the clarifications in the rebuttal.  it addressed most of my concerns. I hope the author will incorporate the rebuttal into the final version.

---

### Official Review · Reviewer_ss93 · 2026-03-03

**Soundness:** 3
**Presentation:** 2
**Significance:** 2
**Originality:** 2
**Overall Recommendation:** 4
**Confidence:** 3

**Summary:**

This paper proposes Deterministic Component Mining (DCM), a three-stage UI-to-code pipeline that separates perception from code synthesis and centers on a portable, component-aware Intermediate Representation (IR). A structure model first predicts a coarse DOM-like tree from a screenshot; a deterministic mining stage canonicalizes this tree and uses hashing/clustering to extract repeated patterns as reusable components with explicit slots and repeat constructs; a framework-conditioned prompting stage then compiles the IR into executable code for different front-end targets (Vanilla HTML, React, Vue, Angular). On the WebCode2M benchmark, the authors report higher CLIP similarity, visual score, compilation success rate, and reusability rate than strong MLLM baselines, with qualitative and human preference evidence.

**Compliance With Llm Reviewing Policy:**

Affirmed.

**Key Questions For Authors:**

Human evaluation: How many samples were evaluated, how were they selected, what was the blinding/randomization procedure, and were significance tests performed? Please report inter-annotator agreement and confidence intervals.

How is CSR measured per framework (especially Angular)? Why is CSR in Angular significantly lower than in other frameworks?

**Limitations:**

The authors have included limitations.

**Strengths And Weaknesses:**

- Strengths

DCM addresses a meaningful gap: producing reusable, component-centric code and transferring across multiple frameworks—an area where many MLLMs struggle per recent benchmarks.

This paper introduces a clear, interpretable IR that explicitly encodes components, repeats, and slot bindings, enabling cross-framework rendering from a single structural artifact. The pipeline is well decomposed and explained; the IR roles and mining steps are described with concrete design choices (vocabulary mapping, masking, hashing).

The experiments are solid to demonstrate that DCM significantly
outperforms baselines on automatic evaluation
metrics and component-level reuse.

- Weaknesses

Human evaluation lacks protocol details (sample size, selection, randomization/blinding specifics, inter-rater agreement, significance testing).

Reusability Rate (RR) is insufficiently operationalized: how is “reusable component structures” detected and scored? Potential metric coupling to DCM’s IR design is a concern.

---

> ### Author Rebuttal · Authors · 2026-03-29
>
> Dear Reviewer ss93,
>
> Thank you for recognizing the value of component-centric, cross-framework UI2Code generation and for the helpful suggestions on evaluation clarity.
>
> ---
>
> **W1: Human-evaluation protocol details**
>
> We agree and will make the human-evaluation protocol explicit in the revision. Specifically, we evaluated `100` pages sampled from the official WebCode2M test set, with roughly balanced coverage across the Short/Mid/Long subsets (`33/34/33`). For each page, we rendered the outputs of DCM and GPT-4o in the same browser environment and presented them side by side in randomized left-right order, with all method identifiers removed. Five professional evaluators independently judged each pair based on two criteria: visual faithfulness to the reference screenshot and engineering quality of the generated code, including structural correctness, component organization, and reusability. Each evaluator selected one of three labels: `Left better`, `Right better`, or `Tie`, and evaluators did not communicate during annotation.
>
> This setup yields `500` total pairwise judgments (`100 × 5`). Aggregated over all judgments, DCM is preferred in `48.0%` of cases, GPT-4o in `17.6%`, and `34.4%` are ties, matching Figure 4. Excluding ties, DCM wins `240` of the `328` non-tied judgments; a two-sided sign test indicates that this advantage is statistically significant (`p < 0.001`). We hope adding these protocol details will make clear that the human study is intended as a reproducible, complementary evaluation of practical frontend quality.
>
> **W2: RR operationalization and possible coupling**
>
> Thank you for pointing this out. To answer the question directly, RR is computed on the final generated code, not on DCM's intermediate representation. For each reference page, we first identify whether it contains repeated visual structures that should naturally be implemented as reusable units (e.g., repeated cards, list items, nav entries). Then, for each generated result in React/Vue/Angular, we check whether those repeated structures are implemented through actual reuse in the output code: namely, a shared reusable component/template that is instantiated multiple times. If the repeated pattern is implemented through such a reusable structure, the sample is counted as positive; if it is rendered by manually duplicated page-specific code, it is counted as negative. RR is the percentage of positive samples among the evaluated samples for that framework.
>
> This is also why we report RR only for component-based frameworks and not for Vanilla HTML in Table 1. RR should be interpreted as an engineering metric that evaluates whether the final code exhibits reusable componentization. The key point is that the metric is applied to the final artifact of every method using the same criterion, rather than reading off DCM-specific IR fields; we will make this distinction explicit and clarify that RR should be interpreted together with CLIP, Visual Score, and CSR rather than in isolation.
>
> **Q1: CSR per framework, especially Angular**
>
> CSR is measured separately for each target framework as the percentage of generated outputs that compile successfully without framework-specific build errors. We agree that this should be stated much more explicitly in the paper, because the current presentation may make the metric look global rather than framework-conditional. In the revision, we will clarify that CSR is computed independently for HTML, React, Vue, and Angular, so the Angular numbers should be interpreted within the Angular setting rather than as a direct reflection of visual quality alone.
>
> Regarding the lower Angular CSR, our reading is that Angular is simply the hardest target in this benchmark for all compared methods. Its stricter template syntax, stronger typing/scaffolding conventions, and higher sensitivity to structural mismatches make end-to-end generation less forgiving than React or Vue. This is therefore a benchmark-wide difficulty pattern rather than a failure mode unique to DCM. Importantly, even in this more difficult setting, DCM remains strongest or near-strongest: on Angular, it improves over GPT-4o from `0.71` to `0.76` on Short, from `0.67` to `0.72` on Mid, and from `0.58` to `0.61` on Long. We will make this framework-specific interpretation explicit in the revision so readers can better understand both why Angular scores are lower overall and why the relative gains of DCM remain meaningful.

---

> > ### Author Rebuttal · Reviewer_ss93 · 2026-03-31
> >
> > Thanks for the clarifications, the rebuttal addressed most of my concerns. Therefore, I will revise my rating.

---

> > > ### Author Response · Authors · 2026-04-01
> > >
> > > We sincerely thank you for raising your score, for your careful review of our work, and for taking the time to revisit it. We are delighted that our rebuttal was able to address your concerns.

---

### Official Review · Reviewer_HbSW · 2026-03-10

**Soundness:** 3
**Presentation:** 3
**Significance:** 3
**Originality:** 3
**Overall Recommendation:** 4
**Confidence:** 3

**Summary:**

The paper attempts to solve the problems of insufficient multi-framework support and a lack of component reusability in the UI2Code generation domain by presenting a neural-symbolic hybrid model called Deterministic Component Mining (DCM). DCM consists of three main modules:

**Vision-to-Structure**: A coarse-level DOM tree in JSON format is predicted from a screenshot using a light-weight structural model.

**Structure-to-Reuse**: Repeating structural patterns are recognized through Merkle-style tree hashing and clustering, resulting in a representation with slot information.

**Reuse-to-Code**: The representation is converted into runnable code in React, Vue, Angular, or native HTML formats using a framework-conditioned prompt.

The experiment results indicate that DCM outperforms the benchmark model by a large margin on visual similarity, compilation success rate (CSR), and reuse rate (RR) on the WebCode2M dataset compared to powerful pre-trained models like GPT-4o and Gemini-2.5-Pro.

**Compliance With Llm Reviewing Policy:**

Affirmed.

**Key Questions For Authors:**

1. If the predicted BBox in Stage 1 has a slight offset, will the hash function H in Stage 2 fail to cluster due to changes in node attributes? Has a fault tolerance mechanism been introduced?

2. When handling nested repeats, while the greedy selection strategy (Priority Score) improves stability, it may sacrifice the extraction of some fine-grained components. Has the author quantified the impact of this sacrifice on code maintainability?

**Limitations:**

Yes

**Strengths And Weaknesses:**

Strengths

1. The "mining-before-generation" paradigm addresses effectively the instability issue of multimodal large models in handling complicated UI structures.
2. Intermediate representations enable the migration of the same design to different frameworks.
3.  The substantial improvement in RR metrics confirms DCM’s ability to successfully detect and merge identical components, such as list items and cards, to produce code that resembles human experts at the engineering level.

Weaknesses

1. The component mining in Stage 2 is highly dependent on the quality of the DOM tree prediction in Stage 1. If the hierarchical structure prediction in Stage 1 fails, so does the component mining in Stage 2.

2. The existing techniques are mostly limited to static UI component reconstruction in terms of visual layouts, providing little support for dynamic interaction logic in UI components.

4. When the visual difference between duplicate components is large, such as in heterogeneous slot content, is the slot alignment algorithm still robust? The robustness evaluation of this long-tail case in this paper is insufficient.

---

> ### Author Rebuttal · Authors · 2026-03-29
>
> Dear Reviewer HbSW,
>
> Thank you for recognizing the value of the mining-before-generation paradigm and the role of the intermediate representation in multi-framework transfer.
>
> ---
>
> **W1: Dependence on Stage 1 and robustness to BBox noise**
>
> Thank you for raising this important point. Our understanding is that Stage 2 is generally robust to ordinary Stage 1 noise, and we will make this more explicit in the revision. The main reason is that the mining stage is intentionally designed around structural invariants rather than raw BBox values. As described in Section 3.4, the subtree fingerprint is computed from canonical node labels and ordered parent-child relations, while geometry and style attributes are deliberately excluded from the hash. Consequently, slight BBox offsets or other small localization noise do not usually change the fingerprint and, in practice, do not materially disrupt clustering.
>
> This robustness is further supported by the canonicalization design before hashing: literal content is masked, tags are mapped into a stable vocabulary, and children are serialized under a canonical order, so matching depends on coarse structure rather than pixel-level coordinates. We have not introduced a separate denoising module for BBoxes because the representation itself already provides substantial tolerance to modest noise.
>
> **W2: Limited support for dynamic interaction logic**
>
> Thank you for pointing this out. Our focus in this work is on a part of UI2Code that is both well-grounded in a single screenshot and highly valuable in practice: recovering a structurally correct, reusable, and executable frontend representation that can be transferred across frameworks. In this respect, DCM contributes an explicit intermediate representation, which enables the system to generate engineering-ready code with a clearer reuse structure than end-to-end monolithic generation.
>
> Dynamic interaction logic is indeed a more challenging layer of the problem, since behaviors such as event handling, state transitions, routing, or asynchronous updates are often weakly observable from a static screenshot alone. We therefore view our current contribution as establishing a strong structural foundation for interactive synthesis rather than attempting to infer latent behavior unreliably. We will clarify this scope more carefully in the revision and emphasize that robust multi-framework structure generation and reusable component abstraction are important prerequisites for richer interactive UI generation in future work.
>
> **W3: Robustness under heterogeneous slot content**
>
> The slot-alignment mechanism is robust when repeated instances share the same structural skeleton and differ mainly in content fields such as text, images, links, or other leaf-level attributes. In that regime, the pipeline works as intended: canonicalization masks literal content in the structure view, preserves generation-relevant values in the content map, and then converts the varying positions across clustered instances into slots plus a data table. This is precisely the setting for common repeated patterns such as product cards, menu entries, or feed items with shared layouts but different content.
>
> Its robustness is more limited when the visual differences are large enough to alter the subtree structure itself. For example, if one card has an extra badge block, a different nested container, or a substantially different internal layout, then exact same-parent fingerprinting will tend not to merge those instances into one cluster. In that case, our method prefers to miss a potential reuse opportunity rather than force an incorrect component abstraction. We view this as a deliberate precision-over-coverage tradeoff: the current design is conservative in long-tail heterogeneous cases.
>
> **Q1: Nested repeats and maintainability tradeoff**
>
> We agree that the greedy nested-repeat selection introduces a real tradeoff. The priority score explicitly favors repeated blocks that are both frequent and structurally substantial, so the algorithm prefers extracting a larger reusable unit over decomposing the same region into multiple smaller components. This usually improves stability in Stage 3 and leads to code that is easier to compile and less fragmented, but it can also suppress some fine-grained components that a human engineer might still choose to factor out.
>
> While we do not report a dedicated maintainability metric in the current submission, the available engineering-oriented results are consistent with this design choice rather than suggesting that it hurts code quality. Relative to GPT-4o, DCM improves the average RR across the 9 component-framework settings (React/Vue/Angular × Short/Mid/Long) from `0.67` to `0.72`, and the average CSR from `0.73` to `0.78`. We therefore view the current evidence as indicating that favoring larger reusable units improves executability and reuse-oriented structure in practice.

---

> > ### Author Rebuttal · Reviewer_HbSW · 2026-04-01
> >
> > Thanks for author's reply. I have no more questions.

---

> > > ### Author Response · Authors · 2026-04-02
> > >
> > > We sincerely thank you for your careful review of our work and for taking the time to revisit it. We are delighted that our rebuttal was able to address your concerns.

---

### Decision · Program_Chairs · 2026-04-30

**Decision:**

Accept (regular)

**Comment:**

This paper proposes DCM, a three-stage UI2Code pipeline that separates structure prediction, deterministic component mining, and framework-conditioned code generation, with the main goal of producing reusable and portable componentized code across HTML, React, Vue, and Angular from the same screenshot.

It has following strengths:

* The problem is good, since current UI2Code systems are still weak on multi-framework generation and reusable component structure.
* The mining-before-generation design is clean, and the IR with repeats and slots makes the pipeline much easier to interpret.
* The empirical results are convincing overall, not just on automatic metrics but also on actual component reuse.

The main concerns were fairly focused: whether Stage 2 is too dependent on Stage 1 structure quality, whether RR and the human evaluation protocol were defined clearly enough, whether the positioning against WAFFLE, Design2Code, and repeated-element evaluation was clear enough, and whether the writing made the motivation and Section 3 harder to follow than necessary.
The rebuttal addressed these points pretty well by clarifying that hashing depends mainly on structure rather than raw BBox noise, adding human-evaluation details, explaining RR as an end-to-end engineering metric on the final code artifact, clarifying the role of WebCode2M and the difference from WAFFLE, and adding extra repeat-coverage style analysis and case studies.

In the end, all reviewers explicitly said their concerns were addressed, and all reviewers agree with acceptance, so I think this paper can be accepted.